# FACT: Learning Governing Abstractions Behind Integer Sequences

**Peter Belcak**[*]
ETH Zürich
8092 Zürich, Switzerland
`belcak@ethz.ch`

**Ard Kastrati**
ETH Zürich
8092 Zürich, Switzerland
`kard@ethz.ch`

**Flavio Schenker**
ETH Zürich
8092 Zürich, Switzerland
`flaviosc@ethz.ch`

**Roger Wattenhofer**
ETH Zürich
8092 Zürich, Switzerland
`wattenhofer@ethz.ch`

## Abstract

Integer sequences are of central importance to the modeling of concepts admitting complete finitary descriptions. We introduce a novel view on the learning of such concepts and lay down a set of benchmarking tasks aimed at conceptual understanding by machine learning models. These tasks indirectly assess model ability to abstract, and challenge them to reason both interpolatively and extrapolatively from the knowledge gained by observing representative examples. To further aid research in knowledge representation and reasoning, we present FACT, the Finitary Abstraction Comprehension Toolkit. The toolkit surrounds a large dataset of integer sequences comprising both organic and synthetic entries, a library for data pre-processing and generation, a set of model performance evaluation tools, and a collection of baseline model implementations, enabling the making of the future advancements with ease.

## 1 Introduction

Ordered lists of integers are the natural representation form for all fundamentally discrete abstractions. These arise when encountering evolutions of discrete-time phenomena, finite symmetries of visual patterns, or algorithmic progressions, where they describe the development of consecutive states of a system, automorphisms of $\mathbb{R}^2$, or program listings, respectively. Sequences of integers are also the representation of choice when linearising structured information for analysis, data compression, and communication, with often-appearing datapoints tending to be encoded in the simplest or shortest form. Testifying to their utility to accurately represent abstractions, completion and extrapolation tasks on integer sequences are a frequent part of general human intelligence and aptitude testing ([42, 31]).

It is the aim and the hope for machine learning models to identify straightforward universal abstractions explaining the training data, rather than to memorise a plethora of small classes of exemplars and interpolate when given previously unseen input. The discovery and internalisation of governing concepts, or simply the learning of underlying rules, thus sits at the centre of artificial intelligence research.

We note that many concepts may be uniquely represented by a sequence of integers naturally (e.g. the squares of the natural numbers determine the polynomial $n^2$; 123, 312, 231, 132, 321, 213 encodes

---

[*]The authors of this work are listed alphabetically.

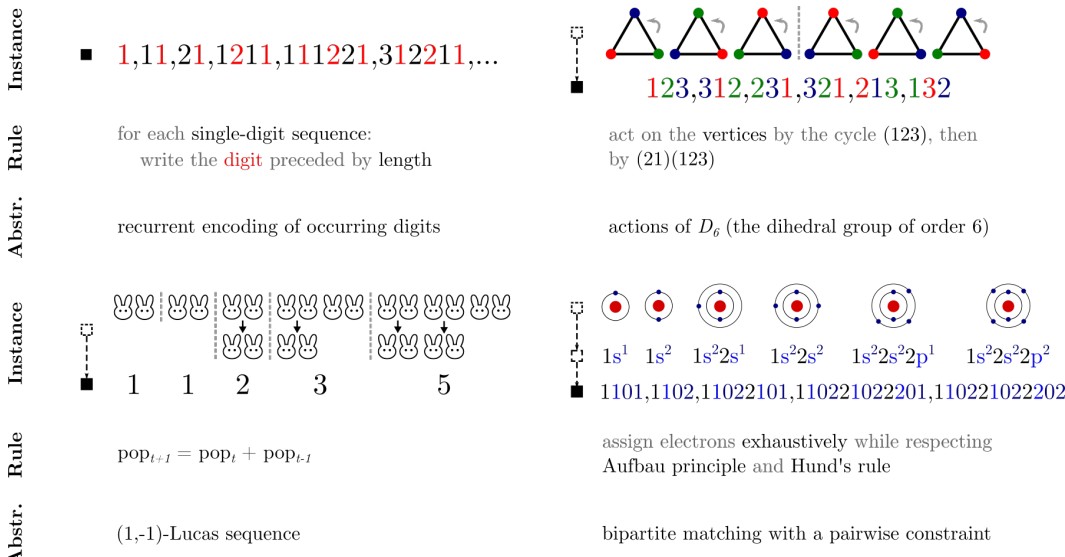

Figure 1: An illustration of an example conceptual learning programme applied to four separate instances: a toy recursive digit-counting sequence, the symmetries of an equilateral triangle, the evolution of an idealised rabbit population as described in [34], and the electron configurations across shells and sub-shells of an atom. In each instance, the raw data is comprehended in its original modality and then turned into an integer sequence. Then, a prior for the rule that applies is formulated. As more samples are observed, the rule is iteratively generalised until the governing abstraction has been fully revealed.

the symmetries of a triangle), while others (such as the rotations of objects in scene) require a continuous space for a proper, scalable description. We recognise integer sequences as a general form for description of concepts completely representable with finite precision (finite in their nature; *finitary*) and put them at the centre of our study.

To discern the learning and understanding of these discrete abstractions from the learning of their representations in various modalities, we introduce a rich dataset of integer sequences together with a compendium of corresponding tasks that are innately related to integer sequences and well-suited to assess the levels of human-like understanding exhibited by machine learning models. Focusing solely on integer sequences, we thus set the concepts being learned apart from virtually all complexity stemming from the learning of a representation, making the learning process less resource-intensive and the interpretation of evaluation results more straightforward. An example learning programme making this distinction and proceeding to high-level abstractions is pictured in Figure 1.

Modern machine learning methods have been shown to posses the ability to comprehend (or at least pattern-match) convoluted concepts appearing in various data modalities, especially by the means of using deep learning to construct informative representations of the entities studied. In spite of working with number sequences, we take a step away from symbolic regression (which has so far dominated the notion of *understanding* in the area) and, tending to the trend, employ instead a multi-faceted approach in which sequences are characterised by their properties and relations to other sequences, rather than by explanatory symbolic formulas that are more readily interpretable by humans. We expand on the relationship of our work to symbolic regression in Appendix F.

Aiming at the comprehension of abstractions behind concrete representations of finitary phenomena, we unlock a new mode for evaluation of the quality of the abstractions learned. A guess, or an estimate, of the rule behind an integer sequence, which further leads to correct predictions of the sequence's elements on previously unseen inputs, is arguably more desirable than an estimate that describes the sequence well only for inputs known in training. The fundamentally algorithmic nature of the problem of learning finitary abstractions thus makes the problem of extrapolative generalisation well-defined, and allows us to consider extrapolative generalisation performance as a criterion for model assessment.

Our contributions are:

- the introduction of a large dataset of integer sequences comprising data from both organic and synthetic sources and curated for subsequent use in tasks challenging models to develop understanding of the concepts determining the data (Section 2, [5]),

- complementing the above, a utility library (FACTLIB [6]) for integer sequence data processing and generation,

- the introduction of a variety of tasks designed to evaluate the model comprehension of conceptual patterns in integer sequences with a clearly established order of difficulty (Section 3),

- a battery of evaluation metrics tailored to the above tasks to appropriately assess model performance and track progress in this sub-area of knowledge representation and reasoning, and

- a collection of baseline models, both classical and neural, implemented to facilitate seamless experimentation (Section 4, [4]).

## 2 Dataset

As a part of FACT, we introduce a dataset consisting of over 3.6 million integer sequences. The structure-giving starting point for the dataset was the data made available by the Online Encyclopedia of Integer Sequences ([39]). The OEIS is an organically grown comprehensive reference on noteworthy sequences of integers, compiled over decades to aid work in mathematical sciences. We have reviewed the OEIS4 data, set apart a suitable subset of 341,000 entries, and processed it specifically for use in machine learning, in line with the license requirements. Each entry of the dataset is now annotated by up to 18 features conveying the information about the nature, properties, and purpose of the sequence. In Figure 4, we give an overview of the result of this processing step. A full discourse on the extensive curation, refining, and automated annotation effort undertaken can be found in Appendix A.

With the encyclopedia entries aimed at a human reader, we observed that many covered their respective categories only very sparsely, relying on the associated natural language descriptions and the human ability to abstract to make the categorical connection. Our initial experiments with the baseline models (cf. Section 4) further confirmed that much of the data did not reach the critical mass of information necessary for reliable use in machine learning applications. We hence systematically extended the dataset by synthetically generated sequence branches while abiding by the structure and nature of the stem encyclopedia entries and providing carefully engineered automatic annotations wherever possible.

### 2.1 Synthetic Generation

Our principal inspirations were the notions of Kolmogorov complexity and Solomonoff probability [33, 20, 41]. Starting on the level of categories (cf. Figure 4), we defined a context-free grammar $\mathcal{G}_c$ for each category $c$, and then used $\mathcal{G}_c$ to generate ever longer formulas, which were in turn used to generate the sequences. This was done abiding by the notion that the growing length of formulas reflects itself in increasing complexity of rules and therefore generated sequences. An example grammar, used for the production of polynomial formulas, is given in Figure 3.

For each category, a total of 500K synthetic sequences have been generated. The length of the formulas used to generate these sequences was continuously increased following a logarithmic schedule, thereby favouring shorter formulas while still ensuring the presence of sequences from longer formulas. We give all details of our generation procedure for each sequence category in Appendix B.

The combined result of the OEIS curation and the dataset extension process is therefore a large dataset seamlessly integrated into FACT and easily extensible by FACTLIB, if required by more complex tasks or larger applications.

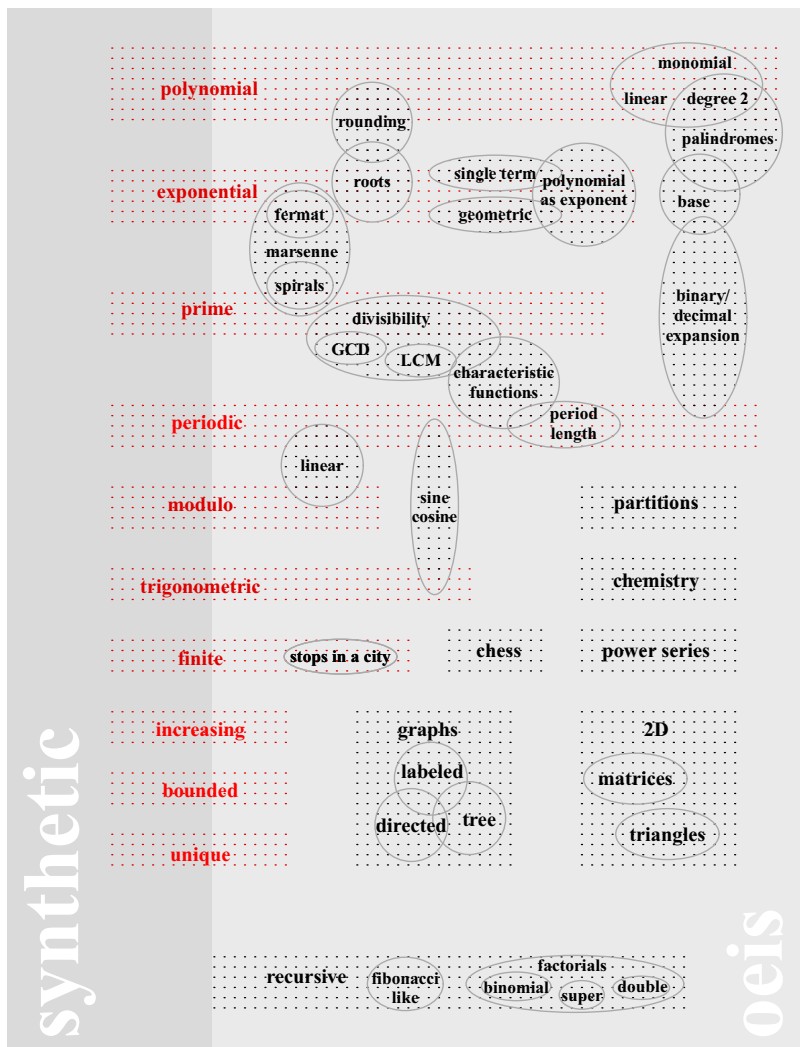

Figure 2: The categories in the FACT dataset. It is composed of synthetic and OEIS entries. Each group in the synthetic part consists of 500,000 sequences, whereas the sizes of the OEIS groups vary. Dotted regions represent the main categories identified in the dataset. Ellipses define the sub-categories from our processing step in OEIS. Red dots mark groups that are augmented with synthetic data (and used in our benchmarking setup).

$$
\begin{aligned}
T &\rightarrow Var \mid Const \\
N &\rightarrow Add \mid Sub \mid Mult \mid Pow \mid NConst \mid T \\
Add &\rightarrow (N + N) \\
Sub &\rightarrow (N - N) \\
Mult &\rightarrow (N * N) \\
Pow &\rightarrow (N ** NConst) \\
NConst &\rightarrow (ConstPos\ NConst) \mid Const \\
Var &\rightarrow x \\
Const &\rightarrow 0 \mid 1 \mid 2 \mid 3 \mid 4 \mid 5 \mid 6 \mid 7 \mid 8 \mid 9 \\
ConstPos &\rightarrow 1 \mid 2 \mid 3 \mid 4 \mid 5 \mid 6 \mid 7 \mid 8 \mid 9
\end{aligned}
$$

Figure 3: The context-free grammar used for the synthesis of formulas used for the generation of Polynomial sequences. $NConst$ denotes a number constant, $T$ denotes a term, and $N$ denotes the root non-terminal for the polynomial expression.

# 3 Benchmark

Based on the dataset of the previous section, we propose a set of benchmarking tasks and evaluation metrics to assess a wide range of methods for their understanding of governing concepts behind evolutions of integer sequences.

## 3.1 Motivation

Consider the initial segment $0, 2, 4, 6, 8, \ldots$ of a sequence and assume we are tasked with proposing reasonable candidates for the number that follows $8$.

How could we go about evaluating the quality of our suggestions?

In the spirit of symbolic regression, we may choose to insist that there must be a single formula that produces the members of the sequence in order. But, even under such condition, for every suggested continuation integer there exists a degree-$8$ polynomial accommodating the continued sequence. This is despite the fact that a human would intuitively be most likely to suggest $10$ as a reasonable continuation, perhaps even justifying it by the observation that the first $5$ elements of the sequence follow the pattern of $(2n)_{n \geq 0}$.

As described in Section 2, we have therefore set the generation methods of the synthetic part of our dataset to explicitly apply the parsimony principle by varying the length of generative formulas to control the complexity of the resulting sequences. Data generated in this manner is guaranteed to encompass sequences coming from rules of varying degrees of complexity by design, rather than by chance.

The focus on preferring shorter rules over longer ones would, however, be too artificial if employed alone. Consider the initial segment $1, 1, 2, 3, 5, \ldots$. While it is tempting to promptly claim that the numbers come from the Fibonacci sequence and that $8$ should follow, an answer arising more naturally in the context of chemistry is $9$, as the continued sequence represents the count of all possible $n$-carbon alkanes.

Hence, striding ahead of the symbolic regression under Occam's razor, we add a processed part of the OEIS dataset into our evaluation procedures as an indicator of the sequences' appeal across a multitude of scientific disciplines. Note that this is an improvement made in addition rather than in contrast to the principles for synthetic sequence generation, as many of the real-world sequences collected in OEIS do indeed obey straightforward rules.

## 3.2 Structure

As motivated in Section 3.1, for each benchmarking task we provide two evaluation sets: one for synthetic data and one for OEIS data. Our benchmark thus pushes for the design of machine learning models that identify simple concepts and rules (evaluation on synthetic data) but still retains enough of cross-domain generality to have practical impact in different disciplines (evaluation on OEIS).

The benchmark consists of tasks with an established order of difficulty over two dimensions: the task type and scope. We distinguish between the tasks of sequence classification, sequence similarity, next sequence part prediction, sequence continuation, and sequence unmasking – each detailed below.

In addition, we perform each task to the extent of two different scopes: within and across categories. The case of performing the task within categories is the simpler of the two setups, since the category from which the sequence originates is known in advance, and this information is thus also available at the time models

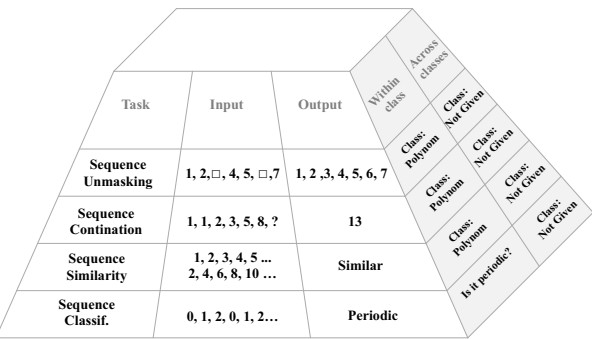

Figure 4: Our tasks ordered by the level of difficulty over two dimensions: type and scope.

are designed. Nevertheless, our baselines (cf. Section 4) are oblivious to this information and instead treat the category scope as if nothing was known about the data.

In our particular benchmarking setup, our dataset is split into four parts, namely the training, validation, synthetic test, and organic test sets. The training and validation sets consist of only synthetic sequences. The size ratios between training, validation, synthetic testing, and OEIS testing datasets are 9:1:1:1. For convenience and future reference, we make this divided-up data available as separate datasets. Nevertheless, the FACT dataset is also available in one piece, allowing users to choose their own splits or supplement their own data generated by FACTLIB, according to their needs.

### 3.3 Task Types

In this section we present 5 types of tasks in order of difficulty, for which we take the upper bound on the difficulty of the task instances comprising the given task. For example: Every instance of sequence continuation belongs to the set of sequence unmasking instances. But, for every instance of sequence continuation (except for the trivial one where we only begin with one number) there is an instance of unmasking that is harder. Such an instance can be formed by further asking to unmask a sequence element somewhere in the initial segment provided for continuation. Hence, the upper bound on the difficulty of sequence unmasking is strictly higher than the upper bound on the difficulty of sequence continuation.

#### 3.3.1 Sequence Classification

The simplest task type in our benchmark is classifying in which category the sequence belongs. The chief goal of this task is to evaluate whether models can distinguish and identify general patterns in sequences, both across and within different categories. Note that category membership may not necessarily be unique. For example, a sequence can be bounded, but also periodic. For this task we use all categories that possess a synthetic counterpart within our dataset. Each sample consists of an array of integer numbers for the given sequence class. We distinguish between two task sub-types and give their objectives:

- **One-vs-Rest (OvR).** *Obj.* to identify whether the sequence belongs to a specified category or not. For this case, we provide a balanced dataset in each category. This is a binary classification task, and as such, we use accuracy as the evaluation metric.

- **Multiclass classification.** *Obj.* to predict, for every sequence, all categories to which it belongs. The performance is measured with the macro average F1 score (i.e. the mean of individual per-class F1 scores) due to inherent imbalances in our dataset.

#### 3.3.2 Sequence Similarity

The similarity task aims to assess model ability to represent sequences in a way that reflects their similarities in type (e.g. agreeing on category membership) or properties (such as being periodic or unbounded) in the spirit of [24]. The objective is to embed sequences into an embedding space such that sequences belonging to the same category are closer to one another than to sequences of categories to whom they do not belong. We evaluate using

- the Recall@$k$ score, where the $k$ candidates for a sequence $s$ are proposed by sampling $k$ sequences from each category and then ordering the category labels according to the distance of the carrier points from $s$; and

- the top-$k$ root mean squared error – the root mean squared error (RMSE) across the top $k$ similarity candidates. Given $k$ predictions of a model $\{\hat{y}_i\}_{i \in \{1,...,k\}}$ and ground truth $y$, we define the top-$k$ RMSE as $\min_{i \in \{1,...,k\}} \mathrm{RMSE}(y, \hat{y}_i)$. In other words, given all generated predictions we report the RMSE of the prediction closest to the ground truth.

Our choice of evaluation metrics is grounded in the observation that sequences generated from similar simple rules often eventually diverge, and that to a large extent. This task generalises sequence classification.

### 3.3.3 Next Sequence-Part Prediction

As seen in natural language processing [10], asking a model to decide whether two sentences follow one another can be beneficial for testing of whether a model understands its inputs. Given two contiguous sub-sequences $s_1$ and $s_2$, the objective of the next sequence-part prediction (NSPP) task is to determine whether the sub-sequence $s_2$ is a valid continuation of $s_1$. We create a balanced dataset for this task, by using all categories of our dataset that have a synthetic counterpart. NSPP is then simply a binary classification task and the performance is measured by prediction accuracy. This task is strictly more difficult than the similarity task, since it demands that the model not only understands the key properties of the sequence but also possesses the ability to discern unlikely or unfeasible combinations of sequence parts from the feasible ones.

### 3.3.4 Sequence Continuation

The fourth task type in our difficulty hierarchy is to suggest the next entry in a given sequence $s$. This task is *extrapolative* in its nature, and is meant to challenge model understanding beyond making a binary decision between externally provided suggestions. As such, this task demands better understanding of the rules governing sequences than next sequence-part prediction – hence it's placement above NSPP in the difficulty hierarchy. We distinguish two sub-types of this task, namely the single-shot and multi-shot continuations, differing only in the number of candidates the model is expected to provide for the continuation. We use the root mean squared logarithmic error (RMSLE) and top-$k$ RMSE for each of the sub-types, respectively.

### 3.3.5 Sequence Element Unmasking

At the apex of our complexity hierarchy is the task of unmasking marked elements of a provided sequence. The sequence continuation task can be viewed as a special case of unmasking, where only the last element is masked. We consider only multi-shot unmasking and choose top-$k$ RMSE as the evaluation metric.

## 4 Baseline model performance

We run extensive experiments on the proposed benchmark as a starting point for further research. We consider classical machine learning methods as well as large neural networks. The experiments in this section highlight the feasibility of learning many different patterns in integer sequences, but also some of the limitations of the existing methods.

### 4.1 Models

To provide baselines for model performance on the above tasks, we use a total of 24 different models across our benchmarking tasks, namely 4 neural models (dense, recurrent, and convolutional networks, and transformers), 9 classical classifiers ($k$-nearest neighbours, Gaussian naive Bayes, linear support vector machine, decision tree, random forest, gradient boosting, AdaBoost, XG Boost, dummy classifier), and 11 standard regressors ($k$-nearest neighbours, linear regressor, ridge regressor, lasso regressor, Elastic Net, single decision tree, random forest, gradient boosting, AdaBoost, dummy regressor).

We give details on their implementations and hyperparameter settings in Appendix C.

### 4.2 Results

A simple overview of the performance of the baseline models can be found in Table 1. A comprehensive, detailed listing of our results, including results of evaluations on the category level, can be found in Appendix D.

### 4.3 Metric Interpretation

The macro-averaged F1-scores for the *sequence classification* task in Table 1 suggest that even the best-performing models have an average score of little over $0.5$. The F1 score of $0.5$ can be achieved

| Model | Dataset | Task | | | | |
|---|---|---|---|---|---|---|
| | | *classification* | *next part pred.* | *continuation* | *similarity* | *unmasking* |
| | | [*F1 score*] | [*binary-accuracy*] | [*RMSLE*] | [*top-5-RMSE*] | |
| MLP | oeis | 0.33 | 0.733 | 0.597 | 0.301 | 2.918 |
| | synth | 0.43 | 0.943 | 0.430 | 1.690 | 3.408 |
| RNN | oeis | *0.37* | *0.869* | 0.603 | 0.383 | 2.944 |
| | synth | **0.53** | **0.984** | 0.406 | 0.438 | 3.379 |
| CNN | oeis | 0.22 | 0.551 | 0.733 | 0.428 | *2.440* |
| | synth | 0.39 | 0.900 | 0.579 | 0.643 | **2.812** |
| Transformer | oeis | 0.33 | 0.736 | *0.578* | *0.267* | 2.811 |
| | synth | 0.44 | 0.938 | **0.395** | **0.270** | 3.091 |
| k-Nearest Neighbours | oeis | 0.33 | – | 0.808 | – | – |
| | synth | 0.41 | – | 0.486 | – | – |
| Gaussian Naive Bayes | oeis | 0.23 | – | – | – | – |
| | synth | 0.37 | – | – | – | – |
| Support Vector Machine | oeis | 0.31 | – | – | – | – |
| | synth | 0.35 | – | – | – | – |
| Decision Tree | oeis | 0.36 | – | 0.730 | – | – |
| | synth | 0.49 | – | 0.427 | – | – |
| Random Forest | oeis | 0.34 | – | 0.730 | – | – |
| | synth | 0.51 | – | 0.427 | – | – |
| Grad.-Boosted Rand. Forest | oeis | 0.27 | – | 0.702 | – | – |
| | synth | 0.40 | – | 0.484 | – | – |
| AdaBoost | oeis | 0.31 | – | 0.842 | – | – |
| | synth | 0.38 | – | 0.662 | – | – |
| XGBoost | oeis | 0.37 | – | 0.719 | – | – |
| | synth | 0.51 | – | 0.433 | – | – |
| Elastic Net Regressor | oeis | – | – | 0.814 | – | – |
| | synth | – | – | 0.716 | – | – |
| Ridge Regressor | oeis | – | – | 0.797 | – | – |
| | synth | – | – | 0.682 | – | – |
| Lasso Regressor | oeis | – | – | 0.827 | – | – |
| | synth | – | – | 0.747 | – | – |
| Linear Regressor | oeis | – | – | 0.797 | – | – |
| | synth | – | – | 0.682 | – | – |
| Dummy | oeis | 0.50 | – | 0.923 | – | – |
| | synth | 0.50 | – | 0.877 | – | – |

Table 1: An overview of the results for all tasks, evaluated across the whole dataset. MLP, RNN, and CNN stand for multi-layer perceptron, recurrent neural network, and convolutional neural network. **Emphasis** and **emphasis** mark the best performing models for the OEIS and synthetic data, respectively. For F1 score and binary accuracy, higher is better. For RMSLE and top-5-RMSE, lower is better.

in many ways, but would for example correspond to a precision-recall performance of $0.5 - 0.5$. In the *next sequence-part prediction*, the RNNs appear to be nearing the perfect accuracy score of $1.0$, though some room for improvement remains to be seen in the case of the organic dataset. The root mean squared logarithmic errors of $0.578$ (best OEIS performance) and $0.395$ (best synthetic performance) that appear in the results for the *continuation* task correspond to uniform difference of logarithms of the same magnitude. In contrast, the two respective worst performances among the baselines models correspond to uniform difference between logarithms of $0.923, 0.877$.

The top-5 root mean squared errors are more straightforward to interpret. One would arrive at RMSE between initial sequence segments $s_1, s_2$ if $s_1$ were larger or smaller than $s_2$ by exactly $5$ in every one of its elements. Top-5-RMSEs of $0.267, 0.270$ for the *sequence similarity* task therefore correspond to mean uniform difference of the same magnitude in every element when comparing the true sequence to the best fit from among the top $5$ results of the similarity search. The same metric is used for the *unmasking* task. In the light of the distribution of the elements of the sequences considered covering

values from $0$ to several million, the performances of baseline models, especially in the similarity task, appear to be remarkably strong.

## 4.4 Comparative Analysis

Reviewing the results of Table 1 and further Appendix D, we note that even just the performance of baseline models on the synthetic dataset is often quite strong in absolute terms. We also notice a general trend among all models to perform better on synthetic data than on the organic OEIS sets. This is not unexpected, since the OEIS data is highly varied and comes from a large variety of sources, whereas the synthetic data is generated according to a strict, uniform procedure, thus having a more regular distribution. Nevertheless, we observe that training on synthetic data alone still yields solid performance on the organic dataset across all models.

In the *classification* task, RNNs and random forests achieve the best results across all categories. Unsurprisingly, RNNs very accurately identify bounded and increasing sequences, while random forests lead for modulo, prime, exponential and trigonometric sets. RNNs also show a consistent lead for the *next sequence-part prediction*. Transformers and CNNs dominate the results for *similarity* under the top-$k$ accuracy, with the exception of organic periodic functions, which are best handled by recurrent networks. Transformers alone lead in the same task when evaluating by top-$k$ RMSLE, and likewise for convolutional networks in the *unmasking* task .

The best performances for the *continuation* task are almost evenly split between recurrent networks and transformers, where RNNs lead for polynomial, exponential, trigonometric, and periodic sequences (transformers being the worst performers). Transformers yield the best results modulo, prime, bounded, increasing, and all sequences together.

We lay out our expectations for model performance on this benchmark, also in relation to human ability, in Appendix E.

## 5 Related Work

There has recently been a noticeable movement in neural network research towards understanding how DNNs learn to abstract. On the side of investigation, traditional architectures were analysed by [21] in terms of the emergence of knowledge across the network, a well-defined metric for the generalisation ability of neural networks was introduced in [11], and a methodology to assess knowledge representation in deep neural networks trained for object recognition in computer vision was proposed in [18]. The increasing interest in the learning of abstractions also prompted the incorporation of relevant inductive biases into deep neural architectures and training curricula, as was seen in concept acquisition [46, 14], with the introduction of the neural state machine [16] in computer vision, and causal abstraction analysis [2, 3, 13] in natural language inference. Further, efforts have already begun to assess the capability of models to perform abstracting visual reasoning [47, 1, 48, 7].

A common, classical, and still challenging instance of abstraction learning in the context of number sequences is the task of *symbolic regression*. A number of genetic programming models and purpose-specific datasets have been proposed in the field in its over 30 years of existence, and a systematising benchmark, SRBench [22], was recently introduced. It combines 130 smaller numerical datasets, both organically grown and synthetically generated, with the PMLB [26], and comes with an evaluation of a range of symbolic regression models using a newly-proposed metric. The inherent focus of the task on interpretability makes it suitable for industrial use but leads to challenges in identification of prevailing generative concepts, as two sequences originating from two distinct instances of the same rule may be best fitted by two formulas completely different in their nature.

Focusing on *integer sequences*, the Online Encyclopedia of Integer Sequences (OEIS) was presented in [39]. The entries of the encyclopedia come from both individual human contributors and automated mechanisms for the invention of "interesting" sequences [8]. It was used for sequence classification combining heuristics and machine learning methods in [45], for the sequence continuation task by fully-connected neural networks in [30], for digit-level sequence term regression to highlight the computational limits of neural networks in [25], and for the learning of mathematical properties of integers for use in natural language processing by training OEIS-sequence embeddings in [32]. The latest version, OEISv4, is the most comprehensive source of annotated information on integer

sequences, containing over 300,000 entries. The dataset has further seen use in the emergent sub-area of deep symbolic regression [23, 29, 9, 19].

Our experience shows that the OEIS data is too sparse and too closely tailored to the needs of human reader to be useful for training of machine learning models for integer sequence comprehension. It can still, however, serve as an interesting proxy of "usefulness" (such as in [8]) in model evaluation when appropriately filtered and pre-processed for that purpose.

# 6  Avenues for Future Work

The carrying advantage of the focus on integer sequences instead of other – potentially richer – input modalities is that we can directly interpret the performance scores of individual models as their ability to comprehend sequence-giving abstractions, and have the confidence that no model performance has been hampered by its insufficient understanding of the input representation. Here, while the models we evaluated appear to show some level of understanding of patterns underlying integer sequences, there is still significant room for improvement, especially in multi-class classification, sequence continuation across all classes, and sequence unmasking.

The results of Section 4 were all produced for a "static" mode of operation, in which all of the query data (e.g. the sequence to classify or a sequence prefix to continue) was provided to the model upfront and as a whole. A mode of operation occurring perhaps more naturally in most practical scenarios is that in which the model is active in its learning and interacts with an oracle, polling for information until it is confident that it can provide and answer. Such interactive variants can be readily formulated for all tasks in Section 3.3, but require a more sophisticated set of evaluation metrics that takes into consideration the amount of information the model requested before producing an answer. We believe that this setup deserves attention as it can provide valuable insights into model reasoning, and we aim to tackle it in our future work.

# 7  Conclusion

Integer sequences frequently arise as the natural representation form for finitary phenomena. Focusing on integer sequences allows us to directly address the problem of learning abstractions and removes the otherwise necessary overhead of learning modality-specific representations.

The benchmarking toolkit for integer sequences presented in this work is by design general in its purpose, aimed at fundamental understanding due to its use of integers as primitive representations, and hierarchically encompasses many of the tasks that have previously appeared isolated in the literature.

It is our hope that our work will help attract attention to the challenges of designing models that perceive logical relationships ruling over the training corpora and reason during inference, thus helping to facilitate future advancements on the frontiers of general artificial intelligence.

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
