# OpenReview forum: "FACT: Learning Governing Abstractions Behind Integer Sequences"
_NeurIPS.cc/2022/Track/Datasets_and_Benchmarks — NeurIPS 2022 Datasets and Benchmarks _

### Official Review · Reviewer_72XJ · 2022-07-02
**A benchmark for the inference of Integer Sequences properties**

**Rating:** 10
**Confidence:** 3
**Correctness:** yes
**Clarity:** yes

**Strengths:**

- A large and comprehensive set of benchmarks
- Thorough categorization of integral sequence properties
- A good starting baseline
- Division of the main goal into subtasks of varying difficulties

**Weaknesses:**

- The provided source-code (https://github.com/FACT-Development-Team/FACT) lacks an appropriate documentation on how to setup and use
- In this same line, there is no instructions on how to extend the benchmark and the ML methods available


**Additional Feedback:**

none

**Documentation:**

yes

**Relation To Prior Work:**

yes

**Summary And Contributions:**

This paper proposes a set of benchmarks with the objective of determining certain properties of integer sequences such as whether they are finite, periodic, etc., identifying similar sequences (w.r.t. to their properties), detecting whether two sequences, s1 and s2, follow each other, identify the next entry of a sequence.

By breaking down the tasks of understanding integer sequences, they can investigate the potential of ML algorithms for each task individually and even stimulate the creation of models specific to one of these tasks. Overall, this could provide new insights for future research revolving this subject, even for symbolic models describing the sequence.

---

> ### Author Response · Authors · 2022-08-23
> **Author-Reviewer Discussion Response**
>
> Dear reviewer,
>
> Many thanks for your review. In response to the cited weaknesses:
>
> > * The provided source-code (https://github.com/FACT-Development-Team/FACT) lacks an appropriate documentation on how to setup and use
> > * In this same line, there is no instructions on how to extend the benchmark and the ML methods available
>
> This has now been addressed. Every part of the repository and dataset is equipped with a README file, key code components now all have docstrings and console interface, and we additionally also provide a QuickStart guide to train a simple transformer-encoder classifier on our benchmarking data. The dataset splits are now clarified in the main text, appendix, the appropriate README files, and also provided as ready-to-use files. The seeds are now specified both in the appendix and the appropriate README files.
>
> The comments on the top level give a comprehensive list of our changes.
>
> Please let us know if you feel there are other items yet to be addressed.

---

### Official Review · Reviewer_Bx7J · 2022-07-20
**Valuable dataset, benchmark, and software for research into learning abstractions.**

**Rating:** 7
**Confidence:** 4

**Strengths:**

The package (dataset, benchmark, software) is well-designed overall and will help advance research in abstract reasoning with machine learning models, a task that is highly relevant to the NeurIPS community.
The software engineering thinking that went into the design of the library is appreciated.

**Weaknesses:**

Given that one of the advertised motivations behind the work is to help advance toward artificial general intelligence (AGI), the absence of few-shot learning scenarios in the benchmark is surprising.
See also "Correctness".

**Additional Feedback:**

- ll. 25–27: "sits at the centre of artificial intelligence research" - you're a bit overselling here, maybe tone it down a bit (e.g., "is a cornerstone" or so, as there can be multiple of those but usually only one center, and – see "Correctness" – you don't address few-shot learning)
- Figure 3: maybe mention the direction of the orderings in the caption (top to bottom and left to right); also, I assume that tasks that don't share a row or a column are incompatible, but you might want to make that clear in the text
- l. 160: "macro average F1 score" – I can guess what "macro" means here, but maybe you can make it a bit more explicit
- Tables 1–3, 6–9 could be more informative as figures
- Figure 4 is missing axis labels
- ll. 483/484: scoring from -2 to 2 would have been more intuitive
- ll. 545–552: You acknowledge that palindromicity is representation-dependent, which makes it stand out among the characteristics you define. Why did you include this anyways?
- ll. 556–562: I found this a bit confusing – e.g., does "primality" in your dataset now mean "consists only of primes" (ll. 541/542) or "is somewhat related to primes" (ll. 558–562)?

Typos and language things (only those that caught my eye while reading):
- l. 4 in the caption of Figure 1: "sub-shells" -> "sub-shells _of an atom_" (?)
- l. 4 in the caption of Figure 1: "is comprehended its original modality" -> "is comprehended _in_ its original modality"
- l. 2 in the caption of Figure 2: "in synthetic part" -> "in _the_ synthetic part"
- l. 2 in the caption of Figure 2: "whereas the size" -> "whereas the size_s_"
- l. 107: comma after "design"
- l. 114: no "the" before "Occam's"
- l. 116: "multitude of disciplines of science" -> "multitude of _scientific disciplines_"
- l. 158: commas before and after "and as such"
- l. 159: commas before and after "for every sequence"; "where it belongs" -> "_to which_ it belongs"
- l. 166: "each other" -> "one another"
- l. 171: "mean square error" -> "mean square_d_ error" (twice)
- l. 180: "for model understanding of its inputs" -> "to test whether a model understands its inputs"
- l. 184: commas before and after "and as such"
- l. 195: comma after "sub-types"
- l. 203: "in" -> "on"
- l. 206: "current limitations of the existing methods" -> "limitations of existing methods"
- l. 209: "bench-marking" -> "benchmarking"
- l. 209: "four" -> "4" (consistency)
- ll. 209/210: "transformer networks" sounds weird, maybe say "dense, recurrent, and convolutional neural networks, and transformers"
- ll. 211/213: "Ada Boost" vs "AdaBoost" - pick a spelling
- l. 218: comma after "our results", "the" before "category", comma after "level"
- l. 252: "the purpose" -> "that purpose"
- l. 261: "periodic functions" -> "periodic functions, which are"
- l. 280: "requires" -> "require"
- l. 282: "model's reasoning" -> "model reasoning"; "and aim" -> "and we aim"
- l. 286: "learning of abstractions" -> "learning abstractions"
- l. 324 (ref. 13): "ofinformation" -> "of information"
- l. 414: "easy to use" -> "easy-to-use"; comma after "example"
- l. 454: comma after "section"
- l. 481: comma after "To this end" (many commas missing in the Appendix after this point, not documented anymore below)
- l. 499: "returns" -> "return"
- l. 539: "withing" -> "within"
- ll. 541, 543, 553: line breaks after first word missing
- "non terminal" -> "non-terminal", globally (relevant for Appendix B)
- l. 623: no full stop after "function"
- l. 655: "for" before "which"
- l. 665: "four" -> "4"
- l. 749: "self contained" -> "self-contained"

**Clarity:**

The paper is structured well and written clearly.

I would prefer figures over tables to present the baseline results (in the main paper and in the Appendix), but I acknowledge that dumping results into tables is common practice.

**Correctness:**

Overall, the dataset was constructed in a sound way, and the evaluation methods and experimental design of the benchmark are appropriate and performed correctly.

But:
- On judging sequence complexity by the length of the generating rule: The rule you use to generate a sequence might not be the only rule generating a sequence. Can you say anything about the length of your generating rule vs. the length of the shortest generating rule in existence?
- On your hierarchy of task difficulty: Sequence Continuation vs. Sequence Element Unmasking - unmasking _one_ element in the middle of a sequence appears (to me) to be easier than unmasking _one_ element at the end of a sequence, so the difficulty relationship you assume here is not entirely intuitive
- On the experimental setup and the reported baseline results: I might have missed something, but are you using a _single_ train-eval-test split? How is that split determined (beyond the details you give in the paper)? It might make sense to deliver _several_ train-eval-test splits with your benchmark, which would allow us to understand how performance varies across splits. In that spirit, I would have expected to see some indicators of variance in the result tables as well.
- In your result tables (if you choose to stick with the tables), make sure you _actually_ highlight _all_ of the best performers (you definitely didn't in Table 1!).
- When presenting results, please specify when higher is better and when lower is better in the captions, especially since you have _both_ cases.
- Similar to ARC (see "Relation to Prior Work"), you might want to consider also providing a _private_ test set against which external users can evaluate.
- What about few-shot learning? If you truly aim at making progress towards AGI, you might want to have a few-shot learning scenario (again similar to ARC), e.g., along the lines of "given a sequence s1, its continuation c1, and a sequence s2, generate the continuation of s2" (or something even more IQ-test-like). Humans don't need many examples to identify the simplest applicable pattern, so one could demand that AGI machines shouldn't, either.

**Documentation:**

All parts of the contribution are documented well overall.

But:
- Please provide additional information on hosting and maintenance in the Appendix of the final version.
- Please make sure that whatever your solution to _present_ the dataset and the benchmark (you mention that you plan to set up a website), you obtain (versioned) DOIs (separately for the dataset, the benchmark, and the library) for the initially published versions – and updated DOIs for all subsequent versions, where applicable.
- Maybe I'm blind, but please make sure you provide a requirements.txt, environment.yml, poetry.lock, or other way to automatically install your software (e.g., distributing via pip or conda).
- More documentation and easy-to-run usage examples for the software library (e.g., Readthedocs, Jupyter notebook tutorials, Colab/Binder) would be appreciated (such as showing a minimal example of how to use the Annotator Class – sure, I can find out from the source code, but well-made examples are usually much faster to understand, lowering barriers to entry).
- It would be good to document the computational resources (time and space) needed to work with FACT effectively in the Appendix.
- Consider using the ML Reproducibility Checklist to document the benchmark (in addition to the Datasheet for Datasets to document the dataset), or at least go through once more to ensure that all content requested therein is documented in some place in your Appendix.
- You might want to license your code separately (with a code-specific license). You could also use the 4.0 version of your CC license, rather than the unported 3.0 one. It is compatible with the OEIS license.
- The datasheet seems to have been filled out unenthusiastically (all the boilerplate, I know), and it should be updated before publication to reflect the situation upon publication (especially the "Uses", "Distribution", and "Maintenance" parts).
- ll. 807–809: Once you have obtained versioned DOIs for the individual parts of your contribution, mention the always-pointing-to-first and the always-pointing-to-latest DOIs in the Appendix.
- ll. 824–825: I recommend using an email address that is not bound to a person's name as a contact address, and setting up email forwarding to the people actually in charge. People move between institutions, and personal email addresses tend to die faster than generic ones.
- The author checklist is missing (it's different from the Datasheet for Datasets!).

**Ethics:**

No.

**Relation To Prior Work:**

The paper discusses relevant related work and specifies clearly how FACT goes beyond the existing literature.

The work reminded me (in some of its spirit) of the Abstraction and Reasoning Corpus (ARC) (https://arxiv.org/abs/1911.01547; https://github.com/fchollet/ARC), which could be mentioned as related work, too.

**Summary And Contributions:**

The authors contribute a large dataset of integer sequences (comprising OEIS sequences and synthetically generated sequences), a benchmark for learning the abstractions governing these sequences, and software tools to work with and extend both the dataset and the benchmark.

---

> ### Author Response · Authors · 2022-08-23
> **Author-Reviewer Discussion Response -- Part 1**
>
> Dear reviewer,
>
> Many thanks for your thorough and detailed review. We believe we have now addressed all points you have raised. Please find a point-by-point response below, and continuation (Part 2) in a separate comment.
>
> > **Correctness**
> >
> > ...
> > But,
> > * On judging sequence complexity by the length of the generating rule: The rule you use to generate a sequence might not be the only rule generating a sequence. Can you say anything about the length of your generating rule vs. the length of the shortest generating rule in existence?
>
> For the synthetic sequences, FACTLIB is motivated from Kolmogorov complexity and generates sequences according to the universal prior: when increasing the length of the rule, the probability of the rule decreases exponentially. The sequences are sampled from universal distribution and as such, sometimes there may exist even shorter rules that generate the exact same sequence. This is done this way by design.
>
> Our aim is to establish a dataset that makes possible experimentation with large neural networks in learning abstractions and representations for sequences. Hence, universal prior introduces a bias towards more simple rules but that does not need to be the shortest one. Note that all deep-learning models are evaluated on input-output behaviour, that is, the deep learning models do not generate the rules themselves. However, we agree that finding the shortest sequence, or other sequences with the same length is an interesting problem on its own. For this reason, we added in README instructions how one can use FACTLIB to generate all sequences for a given length which should make it easier for the community to build studies on top of our framework in this regard as well.
>
> > * On your hierarchy of task difficulty: Sequence Continuation vs. Sequence Element Unmasking - unmasking one element in the middle of a sequence appears (to me) to be easier than unmasking one element at the end of a sequence, so the difficulty relationship you assume here is not entirely intuitive
>
> Your concern is entirely justified, the original description of this was imprecise. We have updated the main text to make it clear we are looking at the upper bounds of the given tasks.
>
> To answer your example in particular: Consider each task as a set of task instances. Every instance of sequence continuation belongs to the set of sequence unmasking instances. But, for every instance of sequence continuation (except for the trivial one where we only begin with one number) there is an instance of unmasking that is harder. Such an instance can be formed by further asking to unmask a number somewhere in the initial segment provided for continuation. Hence, the upper bound on the difficulty of sequence unmasking is strictly higher than the upper bound on the difficulty of sequence continuation. This is the sense in which we look at the "difficulty" of tasks when forming the hierarchy.
>
> > * On the experimental setup and the reported baseline results: I might have missed something, but are you using a single train-eval-test split? How is that split determined (beyond the details you give in the paper)? It might make sense to deliver several train-eval-test splits with your benchmark, which would allow us to understand how performance varies across splits. In that spirit, I would have expected to see some indicators of variance in the result tables as well
>
> *Splits*: We have a 9:1:1:1 split between training, validation, synthetic test, and organic test sets. The text and the datasheet for datasets, as well as the documentation have all been adapted to make this clear. Split data for our benchmark is now also available separately to make reproduction of our results a matter of a few lines in the console.
>
> *Changing splits*: While we could be flexible on the validation proportion and synthetic test proportion, the real limit is given by the size of the subset of the OEIS that can be used for our benchmarking setup. The 1:1 ratio was the natural choice for synthetic test to organic test set size ratio, but one could use more synthetic data if they wanted. We felt that using more synthetic test data could skew the results, so we decided against it. Arbitrary amounts of synthetic test data could be used for training and validation.
>
> *Delivering multiple splits*: Please note that we are not using organic data in training. So, in fact, we are not splitting one dataset into pieces for training and evaluation, but rather using an existing dataset for evaluation, and then supplementing more data in proportion for further evaluation and, of course, training. Hence our decision to go for a single, natural split as above, though with FACTLIB it is very easy to generate more data.
>
> This response continues with Part 2.

---

> ### Author Response · Authors · 2022-08-23
> **Author-Reviewer Discussion Response -- Part 2**
>
> This is the continuation of Part 1 of this response.
>
> Under **Correctness**.
>
> > * In your result tables (if you choose to stick with the tables), make sure you actually highlight all of the best performers (you definitely didn't in Table 1!).
>
> Thank you, we believe we have now fixed all of these, both in Table 1 and in Tables 6-9 of Appendix D.
>
> > * When presenting results, please specify when higher is better and when lower is better in the captions, especially since you have both cases.
>
> Thank you, fixed, both in Table 1 and Tables 6-9 of Appendix D.
>
> > What about few-shot learning? If you truly aim at making progress towards AGI, you might want to have a few-shot learning scenario (again similar to ARC), e.g., along the lines of "given a sequence s1, its continuation c1, and a sequence s2, generate the continuation of s2" (or something even more IQ-test-like). Humans don't need many examples to identify the simplest applicable pattern, so one could demand that AGI machines shouldn't, either.
>
> We concede the lack of few-shot learning evaluative scenarios as a limitation of our current presentation. As we suggest in Section 1, the toolkit presented by our paper is to enable future work in the area rather than cover the area as a whole at this moment. We put FACT forward as a natural starting point, containing data, generation, and curation tools that will jumpstart any inquiries in this direction.
>
> > **Clarity:**
> >
> > The paper is structured well and written clearly.
> > I would prefer figures over tables to present the baseline results (in the main paper and in the Appendix), but I acknowledge that dumping results into tables is common practice.
>
> While figures would indeed be more illustrative of the results, we have decided to go for numbers in tables so that our results can be directly referred to by future work requiring baseline performances.
>
> > The work reminded me (in some of its spirit) of the Abstraction and Reasoning Corpus (ARC) (https://arxiv.org/abs/1911.01547; https://github.com/fchollet/ARC), which could be mentioned as related work, too.
>
> Thank you for the suggestion, we have expanded the Related Work section to also mention the relationship of our work to ARC.
>
> > **Documentation:**
> >
> >  * Please provide additional information on hosting and maintenance in the Appendix of the final version.
>
> Done. Information was also updated in the documentation and the datasheet for datasets.
>
> > * Please make sure that whatever your solution to present the dataset and the benchmark (you mention that you plan to set up a website), you obtain (versioned) DOIs (separately for the dataset, the benchmark, and the library) for the initially published versions – and updated DOIs for all subsequent versions, where applicable.
>
> Done. DOIs are now also the primary mode of reference for the dataset, FACTLIB, and the benchmarking setup.
>
> > * Maybe I'm blind, but please make sure you provide a requirements.txt, environment.yml, poetry.lock, or other way to automatically install your software (e.g., distributing via pip or conda).
>
> Fixed.
>
> > * More documentation and easy-to-run usage examples for the software library (e.g., Readthedocs, Jupyter notebook tutorials, Colab/Binder) would be appreciated (such as showing a minimal example of how to use the Annotator Class – sure, I can find out from the source code, but well-made examples are usually much faster to understand, lowering barriers to entry).
>
> Now addressed in full. Every single part of our code that can be used alone directly or as a library now possesses (1) installation instructions, (2) demonstration instructions, and (3) documentation on where and how to extend it.
>
> *QuickStart Guide*. On top of showcases and starting points in each part of our code, we added a QuickStart guide that uses FACT data for a simple classification task. This QuickStart guide is a Jupyter notebook that can be interactively followed for illustration or used for a head start in specialised work with FACT.
>
> *FACTLIB*. We have added a README file with usage instructions, created a command-line interface for quick prototyping (invoke with --help to see the full list of option), and added docstrings documenting the code behaviour and use to the main parts of the FACTLIB code.
>
> *OEIS Processing*. As with FACTLIB, there is a descriptive README file with (1)-(3) together with an interface to enable users’ own annotations.
>
> *Dataset*. The dataset is now documented and categorised within the ETH Research Collection, and further documented in detail as described under “Dataset” above. The documentation of Appendices A and B was also updated.
>
> This response continues with Part 3.

---

> ### Author Response · Authors · 2022-08-23
> **Author-Reviewer Discussion Response -- Part 3**
>
> This is a continuation of Part 2 of this response.
>
> Under **Documentation**.
>
> > * It would be good to document the computational resources (time and space) needed to work with FACT effectively in the Appendix.
>
> Thank you for the suggestion, we have included summary data in Appendix D.1.
>
> > * Consider using the ML Reproducibility Checklist to document the benchmark (in addition to the Datasheet for Datasets to document the dataset), or at least go through once more to ensure that all content requested therein is documented in some place in your Appendix.
>
> Fixed. For full reproducibility effort changes please also refer to the top-level comment giving the overview of changes.
>
> > * You might want to license your code separately (with a code-specific license). You could also use the 4.0 version of your CC license, rather than the unported 3.0 one. It is compatible with the OEIS license.
>
> Many thanks for the suggestion, the code is now licensed separately.
>
> > * The datasheet seems to have been filled out unenthusiastically (all the boilerplate, I know), and it should be updated before publication to reflect the situation upon publication (especially the "Uses", "Distribution", and "Maintenance" parts).
>
> Fixed.
>
> > * ll. 807–809: Once you have obtained versioned DOIs for the individual parts of your contribution, mention the always-pointing-to-first and the always-pointing-to-latest DOIs in the Appendix.
>
> Fixed (except for always-pointing-to-latest DOI which will be added once a follow-up version has been released).
>
> > * ll. 824–825: I recommend using an email address that is not bound to a person's name as a contact address, and setting up email forwarding to the people actually in charge. People move between institutions, and personal email addresses tend to die faster than generic ones.
>
> Fixed. The new address is fact@ethz.ch and has been filled in everywhere in the text and documentation.
>
> > * The author checklist is missing (it's different from the Datasheet for Datasets!).
>
> Fixed. Now placed directly after references.
>
> > * ll. 25–27: "sits at the centre of artificial intelligence research" - you're a bit overselling here, maybe tone it down a bit (e.g., "is a cornerstone" or so, as there can be multiple of those but usually only one center, and – see "Correctness" – you don't address few-shot learning)
>
> Exactly this phrasing was considered :) and decided against with the following reasoning: they may be multiple things sitting at the centre (perhaps quite a few), but a single missing cornerstone can cause a collapse. If you, however, still advise to change the phrasing we will do that.
>
> >  l. 160: "macro average F1 score" – I can guess what "macro" means here, but maybe you can make it a bit more explicit
>
> Fixed.
>
> > Figure 4 is missing axis labels
>
> Fixed Figure 4 (now Figure 5 due to the inclusion of another figure earlier in the text).
>
> > ll. 483/484: scoring from -2 to 2 would have been more intuitive.
>
> We agree. The scoring scheme is now present in our code, documentation of the curation code, and the dataset itself. We have therefore decided against changing it during this rebuttal for practical reasons.
>
> > ll. 545–552: You acknowledge that palindromicity is representation-dependent, which makes it stand out among the characteristics you define. Why did you include this anyways?
>
> In OEIS as a part of FACT, yes. In the subset of OEIS selected for baseline benchmarking, no, this was not included.
>
> > ll. 556–562: I found this a bit confusing – e.g., does "primality" in your dataset now mean "consists only of primes" (ll. 541/542) or "is somewhat related to primes" (ll. 558–562)?
>
> The latter. In curation, SymPy was used first for symbolic regression and then for testing on primes. We fixed Appendix A to make this clear.
>
> > Typos and language things (only those that caught my eye while reading):
>
> Many thanks for these! Every single one of them has now been fixed.
>
> **Again, thank you for the incredible thoroughness of your review. Please let us know if there are some other items that you feel need to be addressed. Please also refer to the top-level comments for a comprehensive summary of the changes made and feedback implemented.**

---

> > ### Comment · Reviewer_Bx7J · 2022-08-29
> > **Response to Rebuttal**
> >
> > Thank you for addressing the questions I raised in your rebuttal, and for incorporating many of my suggestions in the manuscript, appendix, and online materials; I am glad that you found my comments useful.
> >
> > > Exactly this phrasing was considered :) and decided against with the following reasoning: they may be multiple things sitting at the centre (perhaps quite a few), but a single missing cornerstone can cause a collapse. If you, however, still advise to change the phrasing we will do that.
> >
> > I'm not an empirical linguist but my hunch is that people do not take the visual imagery that literally in their usage of both terms but rather use "cornerstone" to indicate "one of many important things" vs. "center" as "the important thing".
> > If one took the imagery literally, I still think cornerstone would fit better than center (and I have a hard time imagining how multiple things sit at the center – something that "sits" sound like it occupies space that cannot simultaneously be occupied by something else...).
> > There are other potentially less problematic phrases ("has been a guiding theme" or the like), but I am not going to be nitpicking about that.
> >
> > Overall, your work in its current (now also documented!) state will be valuable to the community and I support its publication on this track.

---

### Official Review · Reviewer_GeCe · 2022-07-24
**Review for FACT**

**Rating:** 7
**Confidence:** 4
**Correctness:** Yes.
**Clarity:** Yes.

**Strengths:**

The work introduces an interesting suite of tasks on machines' abstraction ability and number sense. Number abstraction is an important topic that has been barely investigated and it's nice to see the authors working on this challenging domain.

The work is also well-grounded leveraging the OEIS database and structured grammars to generate data.

The work is nicely presented with examples and processing details. And it's good to see the work open-sourced.

**Weaknesses:**

One and maybe the most serious concern of mine is on the goal of this dataset / benchmark. What do you expect out of the machine performance on it? On one hand, we already have various symbolic regression methods that can handle tasks regarding numbers and their relations. If you can have a symbolic regression method solve most of the dataset then what is the point of having it? If it cannot, then why not directly curate another symbolic regression benchmark? On the other hand, I actually suspect if an average person can solve the number sequence problems. We might be familiar with Fibonacci numbers but I'm not sure if an average person can quickly identify the underlying relations. Especially those related to other scientific domains, like the chemistry example in Line 113. If an average person cannot, then shall we expect a machine to be able to?

The manuscript mentions about multi-solutions to a problem. But I'm not sure if I understand it correctly. What if there are equally parsimonious explanations? Say, 1, 3, 5, 7 could be either interpreted as an odd number sequence but also a prime number sequence. Higher-order polynomials are also discussed. How many subsequences (or how long of a sequence) do you think is appropriate to supply to humans to make the distinction?

On the evaluation part, it seems machines are pretty good at it. Is it because machines actually learn the relations in numbers or is it because the dataset is so large that statistical patterns have been observed in the training set? Can you possibly perform tests on generalization to tell the reason? Also, we have recently witnessed the potential of large language models. Can you show some examples that can or cannot be solved by, say, GPT-3, and analyze the results?

There may be incorrect references to Figure 3 in Line 70 and Line 82.

**Additional Feedback:**

See weaknesses.

**Documentation:**

Yes.

**Ethics:**

No.

**Relation To Prior Work:**

The work is highly related to existing abstract reasoning problems in the following. And I think they should be properly discussed.

Raven's Progressive Matrices
[1] Zhang, Chi, et al. "Raven: A dataset for relational and analogical visual reasoning." Proceedings of the IEEE/CVF Conference on Computer Vision and Pattern Recognition. 2019.
[2] Barrett, David, et al. "Measuring abstract reasoning in neural networks." International conference on machine learning. PMLR, 2018.

The Abstraction and Reasoning Corpus
[3] Chollet, François. "On the measure of intelligence." arXiv preprint arXiv:1911.01547 (2019).

Machine Number Sense
[4] Zhang, Wenhe, et al. "Machine number sense: A dataset of visual arithmetic problems for abstract and relational reasoning." Proceedings of the AAAI Conference on Artificial Intelligence. Vol. 34. No. 02. 2020.

**Summary And Contributions:**

The paper proposes a new dataset called FACT, the Finitary Abstraction Comprehension Toolkit. The FACT dataset is composed of integer sequences and challenges a test taker on a variety of tasks, including sequence classification, sequence similarity, next sequence part prediction, sequence continuation, and sequence unmasking. The authors create the dataset by combing the OEIS public repo and designing several grammar structures to extend the dataset. In evaluation, the authors exam both neural and classical classifiers / regressors on the FACT datasets.

---

> ### Author Response · Authors · 2022-08-23
> **Author-Reviewer Discussion Response -- Part 1**
>
> Dear reviewer,
>
> Many thanks for your review. We believe we have now addressed all points you have raised. Please find a point-by-point response below, and continuation (Part 2) in a separate comment.
>
> > **Weaknesses**:
> >
> > One and maybe the most serious concern of mine is on the goal of this dataset / benchmark. What do you expect out of the machine performance on it?
>
> We believe that your concerns have now been addressed. We have added Appendix E to outline expected machine performance, and Appendix F to further clarify on the nature of the relationship of our work to the work on symbolic regression.
>
> > On one hand, we already have various symbolic regression methods that can handle tasks regarding numbers and their relations. If you can have a symbolic regression method solve most of the dataset then what is the point of having it?
>
> Please see the new Appendix F and updated main text that stresses these points. In short, there is little evidence that symbolic regression methods abstract or learn abstractions about their inputs in any way. Although useful in their own right to regress number sequences, they are often powered by search methods and make assumptions on simplicity of the regressed expressions to yield concrete results.
> Neural networks, on the other hand, are commonly associated with the ability to form high-level abstractions in the process of fulfilling their tasks, and currently outperform classical methods in virtually every field of AI previously dominated by the likes of symbolic regression. A very closely related example is the recently published showcase of AlphaCode's power on solving programming puzzles that seemed to have been until then a domain of traditional program synthesis methods.
>
> We have identified the formation of finitary abstractions as a frontier of AI where deep learning is yet to show its strength but where pioneering work is already being made (e.g. Kamienny [9, 19]). The goal of the FACT dataset is thus to allow training and evaluation of deep models (often very data-hungry) that can go beyond regression (as stated in Section 3.1). Otherwise, OEIS alone is a benchmark enough for symbolic regression. An example target model leveraging our dataset would be a large transformer model pre-trained on a subset of our tasks, that can then be readily copied over and fine-tuned for a new task such as sequence complexity prediction or conditional sequence segmentation.
>
> > If it cannot, then why not directly curate another symbolic regression benchmark? On the other hand, I actually suspect if an average person can solve the number sequence problems. We might be familiar with Fibonacci numbers but I'm not sure if an average person can quickly identify the underlying relations. Especially those related to other scientific domains, like the chemistry example in Line 113. If an average person cannot, then shall we expect a machine to be able to?
>
> This we address in the new Appendix E, but please note that this is a rather philosophical question that is not of crucial importance to our goal as stated above. This problem of turning tables in the Turing test was popularised and answered by Penrose [28], and further discussion is likely to become philosophical to an extent beyond the scope of this conference track.
>
> > The manuscript mentions about multi-solutions to a problem. But I'm not sure if I understand it correctly. What if there are equally parsimonious explanations? Say, 1, 3, 5, 7 could be either interpreted as an odd number sequence but also a prime number sequence.
>
> We have modified Sections 3.3.4 and 3.3.5 to better highlight the evaluation in such cases. In short, we use top-k variants of our metrics and phrase the problem of continuing/unmasking such sequences as a retrieval/search task. The brief interpretation section 4.3 and Appendix D with full listing of performances then demonstrate that even the baseline models already lead to reasonable results in such a setting.
>
> > Higher-order polynomials are also discussed. How many subsequences (or how long of a sequence) do you think is appropriate to supply to humans to make the distinction?
>
> We do not take human performance as the guiding criterion in our work, and again refer to Appendix E and [28] as above. The argument of human performance against certain directions of AI research has over the years received what many consider satisfactory answers, and we see it beyond the scope of this conference track to dive deeper into this philosophical issue.
>
> Continued in Part 2.

---

> ### Author Response · Authors · 2022-08-23
> **Author-Reviewer Discussion Response -- Part 2**
>
> This is the continuation of Part 1 of this response.
>
> Under **Weaknesses**.
>
> > On the evaluation part, it seems machines are pretty good at it. Is it because machines actually learn the relations in numbers or is it because the dataset is so large that statistical patterns have been observed in the training set? Can you possibly perform tests on generalization to tell the reason?
>
> This is an excellent point. We have modified the main text (Sections 1, 3.2, and Conclusion) to emphasise the role of OEIS as the ultimate test of generalisation. The OEIS is organically grown and no statistical patterns have been observed within it so far in literature nor in our own curation effort (Appendix A). In fact, it could be considered a large collection of day-to-day outliers. By evaluating all baseline models at OEIS we hope to give more insight into the generalisation performance. As hinted on in Table 1 and thoroughly described in Tables 6-9, the baseline model performance (almost completely regardless of model architecture) on OEIS is comparable with that of performance on the synthetic dataset. This suggests that deep baselines already possess quite some ability to general features across sequences.
>
> > Also, we have recently witnessed the potential of large language models. Can you show some examples that can or cannot be solved by, say, GPT-3, and analyze the results?
>
> Done. Appendix G was added with a simple qualitative analysis of the results returned when prompting GPT-3 Davinci to (a) classify and (b) continue some of the most popular sequences in the OEIS. The prediction of our best baseline is given for comparison.
> One could perhaps hope for a massive, comprehensive evaluation, but please note that GPT-3 offers only a very limited amount of credit for free experimentation.
>
> > There may be incorrect references to Figure 3 in Line 70 and Line 82.
>
> Fixed, thank you!
>
> > **Relation To Prior Work**:
> >
> > The work is highly related to existing abstract reasoning problems in the following. And I think they should be properly discussed.
>
> Thank you for the suggestions, we have updated the Related Work section accordingly and included them in the context of the wider literature. Please see also the updated Section 1 for a clarified description of our effort for *modality-agnostic* abstraction learning (in contrast to work in Computer Vision).
>
> **Please let us know if there are some other items that you feel need to be addressed. Please also refer to the top-level comments for a comprehensive summary of the changes made and feedback implemented.**

---

> > ### Comment · Reviewer_GeCe · 2022-08-29
> > **Response**
> >
> > Thank the authors for providing such detailed explanations. The rebuttal has addressed some of my concerns and I'd like to raise my score.

---

### Official Review · Reviewer_TiHT · 2022-07-26
**New datasets for integer sequence reasoning and abstraction with basic benchmarking across 5 tasks**

**Rating:** 7
**Confidence:** 4

**Strengths:**

**S1**: The paper introduces a new dataset and benchmark for a set of tasks for reasoning over integer sequences. The proposed tasks, such as next-sequence-part-prediction and sequence continuation, are interesting and well-suited for machine learning models.

**S2**: The dataset seems carefully curated and is described well, albeit mostly in the appendix.

**S3**: Various machine learning models are evaluated on the benchmark tasks.


**Weaknesses:**

**W1**: The benchmark implementation seems not usable as-is due to missing documentation (see documentation) and lack of reproducibility. The latter is mainly caused by the lack of fixed seeds or dataset splits for train/test/validation purposes of machine learning models (see correctness).

**W2**: The motivation of this benchmark is, in part, to identify (high-level) abstractions explaining the integer sequences, the benchmark models and results towards that end are not well explored which restricts the value of this paper. Beyond classification/regression performance regarding the reasoning tasks, what insights are machine learning models expected to surface and/or what applications should they facilitate? Which open challenges does the benchmark identify, e.g. from the classification/regression mistakes? No symbolic methods were considered as baselines weakening the motivation for machine learning models. It seems good to illustrate such insights in the experimental section.

**W3**: The main paper does not provide sufficient detail regarding 1) the dataset, 2) interpretation of the baseline and model performances. The dataset contents and preprocessing/feature procedures are covered in the appendix, but its corresponding section in the main paper lacks important details needed to understand it on a high level. Although performances are generated for a large number of models, interpretation is very limited. This benchmark, therefore, reveals limited insights.

**Additional Feedback:**

None.

**Clarity:**

The dataset is explained very (too?) briefly in the main paper, making the high-level construction, preprocessing and annotation of the datasets unclear. The performance comparison is also mainly covered in the appendix, which limits the insights from reading the main paper.

**Correctness:**

The dataset seems to be constructed correctly and detailed preprocessing steps are provided in the appendix (it is strongly recommended to include some of the key details in the main paper). The benchmark results may not be reproducible from the scripts, given that the dataset splits are non-deterministic (random sampled without fixed seed) and not reproducible. The lack of fixed dataset splits for train/test/validation of machine learning models jeopardizes the comparability of future evaluations. The benchmark should not be accepted without fixed seeds or dataset splits. The reported results should also stem from the same dataset splits.


**Documentation:**

Some related datasets are discussed but none of the state-of-the-art approaches were implemented in this benchmark, hence no insights into performance and challenges of the SOTA methods on the presented tasks and datasets are identified. It is also not clear why more conventional symbolic methods were excluded which seems a missed opportunity to motivate the adoption of machine learning models for these tasks. The benchmark, or at least experiments in this paper, would be stronger if such baselines were considered as well.

**Ethics:**

No concerns regarding ethics.

**Relation To Prior Work:**

It is not clear from the documentation (a ReadMe on GitHub) how one could use the GitHub repository to benchmark their machine learning models using the provided code and data. Clear guidance describing which functions to call etc., are needed to make the benchmark usable. One way of implementing this is by writing a readme, an example could be the AutoML benchmark: https://github.com/openml/automlbenchmark#readme, https://openml.github.io/automlbenchmark/index.html. Details for reproducing the experimental results as reported in the paper should be provided as well.


**Summary And Contributions:**

**Update after rebuttal**: the authors did a good job in addressing the main shortcomings of this paper and benchmark, in my opinion. I have increased my score from 5 to 7 based on the revised paper, dataset and documentation.

----

The paper presents a dataset and 5 benchmarking tasks for learned abstraction and reasoning over integer sequences. The dataset comprises real and synthetic sequences and presents 5 tasks, being: classification, next-sequence-part-prediction, continuation, similarity and unmasking. A performance comparison of multiple classic and neural machine learning models is presented.

---

> ### Author Response · Authors · 2022-08-23
> **Author-Reviewer Discussion Response -- Part 1**
>
> Dear reviewer,
>
> Many thanks for your review. We believe we have now addressed all points you have raised. Please find a point-by-point response below, and continuation (Part 2) in a separate comment.
>
> > **Strengths**
> >
> > ...
> >
> > **S2**: The dataset seems carefully curated and is described well, albeit mostly in the appendix.
>
> This has now been addressed. A high-level description of the dataset building process has been elevated from the Appendix to the main text, as much as the space constraints allowed it.
>
> > **Weaknesses**
> >
> > **W1**: The benchmark implementation seems not usable as-is due to missing documentation (see documentation) and lack of reproducibility. The latter is mainly caused by the lack of fixed seeds or dataset splits for train/test/validation purposes of machine learning models (see correctness).
>
> This has now been addressed in full. Every part of the repository and dataset is equipped with a README file, key code components now all have docstrings and console interface, and we additionally also provide a QuickStart guide to train a simple transformer-encoder classifier on our benchmarking data. The dataset splits are now clarified in the main text, Appendix, and the appropriate README files. The seeds are now specified both in the Appendix and the appropriate README files. The split dataset is also available for direct use (references both in text and in READMEs).
>
> > **W2**: The motivation of this benchmark is, in part, to identify (high-level) abstractions explaining the integer sequences, the benchmark models and results towards that end are not well explored which restricts the value of this paper. Beyond classification/regression performance regarding the reasoning tasks, what insights are machine learning models expected to surface and/or what applications should they facilitate? Which open challenges does the benchmark identify, e.g. from the classification/regression mistakes? No symbolic methods were considered as baselines weakening the motivation for machine learning models. It seems good to illustrate such insights in the experimental section.
>
> We believe this has now been addressed. We extended the Baseline model performance section (4) to give interpretation of the results as well as a comparative analysis across models. Further, we have added two new appendices: Appendix E discusses the expectations on model performance and challenges we have identified in the process of performing all the baseline experiments. Appendix F goes into more detail explaining the relationship of our work to that on symbolic regression. We have also updated the main text to further emphasise the differences in motivation.
>
> > **W3**: The main paper does not provide sufficient detail regarding 1) the dataset, 2) interpretation of the baseline and model performances. The dataset contents and preprocessing/feature procedures are covered in the appendix, but its corresponding section in the main paper lacks important details needed to understand it on a high level. Although performances are generated for a large number of models, interpretation is very limited. This benchmark, therefore, reveals limited insights.
>
> This has now been addressed: A high-level description of the dataset building process has been elevated from the Appendix to the main text. Further, Section 4 now has two subsections: one giving direct interpretations of the summary baseline model results, the other one commenting on individual performances of models. Appendix E then gives expectations with respect to model performance.
>
> > **Correctness**
> >
> > The dataset seems to be constructed correctly and detailed preprocessing steps are provided in the appendix (it is strongly recommended to include some of the key details in the main paper). The benchmark results may not be reproducible from the scripts, given that the dataset splits are non-deterministic (random sampled without fixed seed) and not reproducible. The lack of fixed dataset splits for train/test/validation of machine learning models jeopardizes the comparability of future evaluations. The benchmark should not be accepted without fixed seeds or dataset splits. The reported results should also stem from the same dataset splits.
>
> As noted above, we believe that this feedback has been implemented in full. The main text now contains the information requested, the splits and seeds have been clarified in the text and documentation. A split version of the dataset for direct use is available separately.
>
> > **Clarity**:
> >
> > The dataset is explained very (too?) briefly in the main paper, making the high-level construction, preprocessing and annotation of the datasets unclear. The performance comparison is also mainly covered in the appendix, which limits the insights from reading the main paper.
>
> As above.
>
> This response is continued in Part 2.

---

> ### Author Response · Authors · 2022-08-23
> **Author-Reviewer Discussion Response -- Part 2**
>
> This is a continuation of Part 1 of this discussion response.
>
> > **Relation To Prior Work**:
> >
> > It is not clear from the documentation (a ReadMe on GitHub) how one could use the GitHub repository to benchmark their machine learning models using the provided code and data. Clear guidance describing which functions to call etc., are needed to make the benchmark usable. One way of implementing this is by writing a readme, an example could be the AutoML benchmark: https://github.com/openml/automlbenchmark#readme, https://openml.github.io/automlbenchmark/index.html. Details for reproducing the experimental results as reported in the paper should be provided as well.
>
> As noted in Part 2 and in the general response, we believe this too has now been addressed in full. Further, the README under Benchmark gives a step-by-step guide to the reproduction of all our results and implementation of new models. Even further, the QuickStart Guide (newly added) demonstrates step-by-step how to use FACT from first principles with PyTorch. This is in addition to the benchmarking setup we provide in TensorFlow.
>
> > **Documentation**:
> >
> > Some related datasets are discussed but none of the state-of-the-art approaches were implemented in this benchmark, hence no insights into performance and challenges of the SOTA methods on the presented tasks and datasets are identified. It is also not clear why more conventional symbolic methods were excluded which seems a missed opportunity to motivate the adoption of machine learning models for these tasks. The benchmark, or at least experiments in this paper, would be stronger if such baselines were considered as well.
>
> It is our aim to present a dataset and give baseline performances of a handful of models to identify the ballpark for this newly introduced data and tasks. Though valuable in its own right, it is not our aim (and we felt it inappropriate, given the nature of our paper) to review the current State-Of-The-Art in symbolic regression. Please also note that reviewer GVg5 actually seems to have felt that even the models we considered as baselines might have already been too powerful (we discuss this in the response to their review and in Appendix E). Further, SOTA models would be hard to interpret as *base*lines, since they are carefully tailored to variants of our unmasking and continuation tasks. As such, they might be contenders to future models proposed for this benchmark, but are not really baselines in their own right.
>
> In terms of results alone: On a first look, some symbolic regression state-of-the-art approaches could be used for 2 out of 5 tasks considered by us, albeit not readily as the results of the regression would have to be further automatically analysed, with a separate analysis program necessary for each approach.
> Please note, however, that the sheer size of our benchmarking dataset would make the use of most symbolic regression methods infeasible. Compare also to the sizes of the classical benchmarks such as Feynman Symbolic Regression Database (4 GiB) or ODE-Strogatz (a few MB), which are collated under diverse-feature benchmarking collections such as PMLB [26] and SRBench [22].
>
> In terms of motivation: Many classical symbolic regression methods search spaces of valid expressions and fit symbolic representations to the data provided. As such, we felt that their inclusion in a work concerned with abstraction rather than regression or numerical sequences alone would be inappropriate and would shift the general message of our effort. Our message in this work is (oversimplified): “Neural networks are often thought to abstract and build encompassing representations. Here is a toolkit for investigation of precisely that in a *modality-agnostic* setting.” It is not immediately obvious how symbolic regression SOTA methods would play a part in such investigations.
>
> In terms of correctness and validity of results: Many methods used for symbolic regression make precisely the assumptions of simplicity of the underlying generative rule we make in our generative process, which would render their evaluation as a baseline ill-posed.
>
> We hoped to make this clear in Sections 1, 3.1, 6, and Appendix F, and have made changes to the main text to further highlight these differences.
>
> **Please let us know if there are some other items that you feel need to be addressed. Please also refer to the top-level comments for a comprehensive summary of the changes made and feedback implemented.**

---

> > ### Comment · Reviewer_TiHT · 2022-08-27
> > **Clarifying rebuttal and good revision of paper and documentation**
> >
> > Dear authors,
> >
> > I appreciate your thorough reflection on the review(s) and the revised paper, dataset, and documentation. I believe your contributions as-is will be valuable to the community, and, therefore, I have increased my rating.

---

### Official Review · Reviewer_GVg5 · 2022-07-28
**Interesting line of research**

**Rating:** 6
**Confidence:** 4
**Clarity:** The paper is well written

**Strengths:**

* The datasets is well-curated and contains a large number of tasks
* A wide range of baselines is provided and evaluated on the dataset

**Weaknesses:**

* Baseline models already achieve reasonably strong performance, which may limit the impact of the dataset
* The experiments should include few-shot learning scenarios for completeness and better assessment of the relevance of the dataset
* The paper does not report human performance

**Additional Feedback:**

No additional feedback

**Correctness:**

The claims in the paper are correct and the dataset is constructed in a sound way

**Documentation:**

The paper provides details on how the dataset has been collected and the dataset is well documented.

**Ethics:**

There is no ethical concern related to this dataset

**Relation To Prior Work:**

To the best of my knowledge, the paper provides a good discussion of related work in its area

**Summary And Contributions:**

The paper introduces a dataset of integer sequences comprising data from both real-world and synthetic sources. It additionally provides a library named FACTLIB to process and generate additional data. The dataset includes several types of task, namely sequence classification, sequence similarity, next sequence-part prediction and sequence continuation. For each task a performance metric is given.

---

> ### Author Response · Authors · 2022-08-23
> **Author-Reviewer Discussion Response**
>
> Dear reviewer,
>
> Many thanks for your review. We believe we have now addressed all points you have raised. Please find a point-by-point response below.
>
> > **Weaknesses**
> >
> > Baseline models already achieve reasonably strong performance, which may limit the impact of the dataset.
>
> We believe that there is still significant room for improvement in terms of model performance on the tasks we present. In particular, we see that many (if not all) neural baseline models currently considered could benefit from being trained on more than just the one task at which they are then tested.
>
> > The experiments should include few-shot learning scenarios for completeness and better assessment of the relevance of the dataset.
>
> We concede the lack of few-shot learning evaluative scenarios as a limitation of our current presentation. As we emphasise in Section 1, the toolkit presented by our paper is meant to enable future work in the area rather than cover the area as a whole at this moment. We put FACT forward as a natural starting point, containing data, generation, and curation tools that will jumpstart any inquiries in this direction.
>
> > The paper does not report human performance
>
> We have added Appendix E detailing expectations on model performance, also in the context of human performance. Please note, however, that in this case human performance does not give a natural target or baseline as it may in other sub-fields of machine learning. Doing so here leads to the dilemma of "reverse Turing test" as recently popularised (and answered) by Penrose [28]. We refrain from delving into philosophy and choose to align ourselves with Penrose's analysis.
>
> **Please let us know if there are some other items that you feel need to be addressed. Please also refer to the top level comments for a comprehensive summary of the changes made and feedback implemented.**

---

### Official Review · Reviewer_28cY · 2022-07-29

**Rating:** 4
**Confidence:** 4
**Correctness:** Yes

**Strengths:**

Integer sequencing is applicable to many domains from data compression to communication to biochemistry. The authors introduced FACT - a dataset of an integer sequence dataset consisting of data from both organic and synthetic sources. FACT comprises of processed and generated data from OEIS. FACT provides an easier/better/cleaner access to OEIS.

**Weaknesses:**

The two novel contributions of the paper seem to be:
1) Processing and generating synthetic data from OEIS.
2) Designing different task types including sequence classification, sequence similarity, next sequence-part prediction, sequence continuation, and sequence element unmasking.

However, these two novel contributions are not very significant. FACT comprises of 341,000 processed entries from OEIS. The authors then extended the dataset by synthetically generating sequence branches. However, the author did not include details on how the data is generated (refer to section below). And processing an existing dataset is not a significant contribution.

The five different task types are common in many sequence modeling domains such as language. They have also been used in and are popular in language processing, e.g. predicting the next word in a sentence. Thus, the five different task types are not completely new.



**Additional Feedback:**

N/A

**Clarity:**

- “Our initial experiments with the baseline models (cf. Section 4) further confirmed that much of the data did not reach the critical mass of  information necessary for reliable use in machine learning applications. We hence systematically extended the dataset by synthetically generated sequence branches while abiding by the structure and nature of the stem encyclopedia entries and providing carefully engineered automatic annotations wherever possible. ”
    - Examples of data generation?
    - How much of the data was generated/extended
- Synthetic data generation summary needs more details:
    - How exactly is it generated
- “In this section we present 5 types of tasks in order of difficulty. For each task, our dataset is split into three parts: training, validation, and test set. The training set consists of only synthetic sequences, the validation and test sets consist of a combination of samples from the OEIS dataset and the synthetic dataset.”
    - Why not include some OEIS in the training as well?
- In Table 1, “we also notice a general trend among all models to perform better on synthetic data than on the organic OESI sets”
    - What does this say about the quality of synthetic data?

**Documentation:**

Some more detailed documentation of the GitHub repo would nice!

**Relation To Prior Work:**

Yes

**Summary And Contributions:**

- FACT - a dataset of an integer sequence dataset consisting of data from both organic and synthetic sources
- a variety of tasks designed to evaluate integer sequence models
- a collection of the performance of different baseline models on the FACT

---

> ### Author Response · Authors · 2022-08-23
> **Author-Reviewer Discussion Response -- Part 1**
>
> Dear reviewer,
>
> Many thanks for your review. We believe we have now addressed all points you have raised. Please find a point-by-point response below and in Part 2 of this response.
>
> > **Weaknesses**
> >
> > The two novel contributions of the paper seem to be: ... However, these two novel contributions are not very significant.
>
> The significance of a dataset paper is unlikely to be proven until after its publication, when the dataset has been employed in machine learning practice. Nevertheless, we would like to point to the wide-spread use of comparable but not similar benchmarks such as SRBench [22], as well as that of the OEIS dataset [39]. Their popularity within their respective sub-fields indicates that there is appetite among the scientific community for data that can be readily used for training and evaluation of models concerned with abstract arithmetic and numerical understanding.
>
> > FACT comprises of 341,000 processed entries from OEIS. The authors then extended the dataset ... . However, the author did not include details on how the data is generated (refer to section below).
>
> Due to the page limit, in our initial submission we pushed the explanation of the dataset processing and synthetic generation entirely into the Appendix B and decided to focus on other aspects of the paper in the main text. As suggested by other reviewers, we agree that the main ideas of the generation process should form a part of the main text, and the most recent version now reflects this change. An explanation going into more detail can still be found in the appendix.
>
> > And processing an existing dataset is not a significant contribution.
>
> It is our expressed view that the main contribution in this work is not the part concerned with the processing of OEIS, although it forms a contribution on its own. Indeed, it is not the aim of this paper to present novel techniques in processing the existing integer collection. Instead, our focus lies with the creation of a resource, inspired by and formed around the OEIS, for the learning of finitary abstractions by machine learning and more directly deep learning methods. As explained in the main text, the OEIS as it stands today is not sufficient for the accomplishment of this end on its own.
>
> To reiterate for more clarity in the distinction, our contributions are:
>  * the introduction of a large dataset (an order of magnitude larger than OEIS)  of integer sequences comprising data from both organic and synthetic sources and curated for subsequent use in deep learning;
>  * complementing the above, a utility library (FACTLIB) for integer sequence data processing and generation – permitting further generation, should there  be the need;
>  * the introduction of a variety of tasks designed to evaluate the model comprehension of conceptual patterns in integer sequences with a clearly established order of difficulty;
>  * a battery of evaluation metrics tailored to the above tasks to appropriately assess model performance and track progress in this sub-area of knowledge representation and reasoning, and
> a collection of baseline models, both classical and neural, implemented to facilitate seamless experimentation.
>
> > The five different task types are common in many sequence modeling domains such as language. They have also been used in and are popular in language processing, e.g. predicting the next word in a sentence. Thus, the five different task types are not completely new.
>
> It is no coincidence that the tasks we present appear in NLP (and CV), and we do give the appropriate credit in Section 3. In fact, as we say in the Introduction, these tasks even appear in IQ tests.
>
> Most of these tasks, if not all, arise naturally when one wishes to test model understanding of an underlying abstraction. In NLP, the abstraction whose understanding by a model is being tested by unmasking, continuation, or classification, is the information contained in a piece of text. In computer vision, the abstraction is the scene depicted. In our case, the abstraction is the algorithmic (or, more broadly, finitary) rule behind a sequence.
>
> In sum, we only introduce these tasks in the context of finitary abstraction learning. We do not make the claim that these tasks are “new” in any way. If anything, we hope to point out by our work that these tasks are somehow fundamental to learning and testing of human-like understanding.
>
> We modified the main text to clarify this in the light of the point you raise.
>
> > **Clarity**
> >
> > ...
> > Examples of data generation?
>
> As noted above, we have updated the main text to contain details and example figure of data generation. Please also refer to Appendix B for more detail.
>
> > How much of the data was generated/extended
>
> We generated 500 thousand sequences for each group described in the paper. Please refer to the Appendix B (and the updated main text).
>
> > Synthetic data generation summary needs more details: How exactly is it generated
>
> As above.

---

> ### Author Response · Authors · 2022-08-23
> **Author-Reviewer Discussion Response -- Part 2**
>
> This is the continuation of Part 1 of our response.
>
> Under **Clarity**.
>
> > Why not include some OEIS in the training as well?
>
> In our particular benchmarking setup, the main reason for this decision was that the amount of sequences provided in OEIS is very limited – insufficient, even. Furthermore, our initial experimentation indicated that the OEIS dataset is simply too organic and too varied in its nature to be of real use to what are generally “data-hungry” deep learning models. Nevertheless, both the split data and full data are available as separate datasets, and the users are free to choose splits according to their need.
>
> > In Table 1, “we also notice a general trend among all models to perform better on synthetic data than on the organic OESI sets”. What does this say about the quality of synthetic data?
>
> The OEIS data is highly varied and comes from a large variety of sources, whereas the synthetic data is generated according to a strict, uniform procedure, thus having a more regular distribution. This alone does not say much about the quality of the synthetic data, since we observe that training on synthetic data only still yields solid performance on the organic dataset across all models. The main text was also updated to clarify this.
>
> Our synthetic generation procedure could certainly be improved on incrementally to match the OEIS distribution even better. To this end, we provide our users with FACTLIB, which can be directly used to generate more sequences according to new (or just modified) grammars.
>
> > **Documentation**
> >
> > Some more detailed documentation of the GitHub repo would nice!
>
> This has now been addressed. Every part of the repository and dataset is equipped with a README file, key code components now all have docstrings and console interface, and we additionally also provide a QuickStart guide to train a simple transformer-encoder classifier on our benchmarking data.
>
> **Please let us know if there are some other items that you feel need to be addressed. Please also refer to the top level comments for a comprehensive summary of the changes made and feedback implemented.**

---

### Author Response · Authors · 2022-08-23
**Discussion Changes and Implementation of Feedback: Dataset, Documentation, Reproducibility, and Distribution**

In response to the reviewers' comments, we have made a number of modifications and additions to the dataset, FACTLIB, and benchmarking code. Please find an overview of these changes and highlighting clarifications below; we also respond in detail to each reviewer individually.

## Dataset

**Author statement.** An explicit author statement, together with parallel statements in the main text, author checklist, dataset for datasheets, and code and data documentation, has been added as Appendix I.

**Hosting.** The dataset is now available through the ETH Research Collection (DOI 10.3929/ethz-b-000562705). The dataset will be hosted with the ETH Research Collection under the reference for a minimum of 10 years.

**Licensing.** Licences are comprehensively listed in the dataset documentation, code documentation, appendix, and the datasheet for datasets.

**Maintenance.** The maintenance plan and responsibilities are split between the ETH Research Collection and ETH TIK Institute. This has also been reflected in the datasheet for datasets. A new email address (fact@ethz.ch) has been created specifically for queries about the dataset.

**Splits.** A split version of the dataset (split as in the benchmarking setup) is now also directly available through the ETH Research Collection.

**Documentation.** The dataset details are now documented in the main text (Sections 2 and 2.1), Appendix A, Appendix B, datasheet for datasets, with a direct README, and under `FACT/OEIS Processing`.


## Strong, Complete Documentation

Every single part of our code that can be used alone directly or as a library now possesses (1) installation instructions, (2) demonstration instructions, and (3) documentation on where and how to extend it.

**QuickStart Guide.** On top of showcases and starting points in each part of our code, we added a QuickStart guide that uses FACT data for a simple classification task. This QuickStart guide is a Jupyter notebook that can be interactively followed for illustration or used for a head start in specialised work with FACT.

**FACTLIB.** We have added a README file with usage instructions, created a command-line interface for quick prototyping (invoke with `--help` to see the full list of option), and added docstrings documenting the code behaviour and use to the main parts of the FACTLIB code.

**OEIS Processing.** As with FACTLIB, there is a descriptive README file with (1)-(3) together with an interface to enable users’ own annotations.

**Dataset.** The dataset is now documented and categorised within the ETH Research Collection, and further documented in detail as described under “Dataset” above. The documentation of Appendices A and B was updated to reflect the reviewers’ questions.


## Full, Effortless Reproducibility

The benchmarking code has a step-by-step guide for reproduction of our results, available in the README file.

**Seeds and Splits.** Random seeds used and dataset splits are included as a part of the supplementary materials and in the datasheet for datasets. They are also mentioned in README files where relevant. The split dataset is available through the ETH Research Collection.

**Author Checklist.** Author checklist has been updated and re-included into the main text.


## Distribution

The dataset, FACTLIB library, and benchmarking code are distributed through the ETH Research Collection. Each one of them has now been assigned a separate DOI that is also linked in the main text through references, in the datasheet for datasets, and in the documentation files.


FACTLIB is available for use both as a library and for ease of reproduction of data also via a console interface. This has been properly documented, and the users can also use the `--help` option directly with the program.

---

### Author Response · Authors · 2022-08-23
**Discussion Changes and Implementation of Feedback: Main Text and Appendices**

In response to the reviewers' comments, we have made a number of modifications and additions to the main text and supplementary material. Please find an overview of these changes and highlighting clarifications below; we also respond in detail to each reviewer individually.

## Main Text

**Synthetic Data.** Utilising the 10th page, the main text now contains a more detailed description of the synthetic generation process together with an example template grammar (Section 2.1). Appendix B has also been corrected and expanded on as per the suggestions of some reviewers.

**Interpretation of Results.** Section 4 now contains direct interpretations of the summary baseline model results. Note that more results are given in Appendix D, and that details about individual architectures employed for the baselines are available as Appendix C.

**Expected/Target Model Performance.** Section 4.3 indicates some expected performance in terms of metric boundaries, and Appendix E discusses expected model performance also in the context of the point raised by one of the reviewers.

**Comments on Human Performance.** Both the main text and Appendix E now contain further comments on human performance in relation to expected benchmarking performance.

**Goal of the Benchmark.** More emphasis has been put on our goals and ambitions for use of our dataset in the main text. The goals are also discussed in context of the “reverse Turing test” in Appendix E and separately in response to the reviewer.

**Relationship to Symbolic Regression.** A new appendix, Appendix F, has been added to discuss the relationship of our work to symbolic regression – on top of the discussion already outlined (and further clarified and emphasised) in the main text. We also respond to the concrete points raised by the reviewer.

**Related Work Expansion.** We have expanded the related work section to also include related work on abstraction in computer vision.

**Few-Shot Learning.** Our paper in its present scope does not encompass any study on few-shot learning. As we emphasise in Section 1, the toolkit presented by our paper is meant to enable future work in the area rather than cover the area as a whole at this moment. We concede the lack of few-shot learning evaluative scenarios as a limitation of our current presentation. We put FACT forward as a natural starting point, containing data, generation, and curation tools that will jumpstart any inquiries in this direction.

**Typos.** All typo fixes and stylistic suggestions raised by the reviewers have been implemented. Further passes have been made to minimise the occurrence of these.


## Baseline Computational Resources

Appendix D.1 was added together with a table describing a summary of the demand of individual baseline models on the computational resources.


## An Evaluation of GPT-3 Davinci

Appendix G was added with a simple qualitative analysis of the results returned when prompting GPT-3 Davinci to (a) classify and (b) continue some of the most popular sequences in the OEIS. The prediction of our best baselines is given for comparison.

One could hope for a massive, comprehensive evaluation, but please note that GPT-3 offers only a very limited amount of credit for free experimentation.


## Summary of Changes to Appendices

* Appendix B.1 - added; describes the hosting and maintenance of the dataset on top of the information provided in the datasheet for datasets
* Appendix C - modified; to the best of our knowledge, it contains all information needed for the reproduction of our results.
* Appendix C.4 - added; describes the hosting and maintenance of the baseline models on top of the information provided in the datasheet for datasets
* Appendix D.1 - added; gives the breakdown of the baseline model compute time
* Appendix E - added; gives expectations of model performance on the FACT dataset
* Appendix F - added; on top of the information already provided in the main text, addresses some specific points of the relationship of our work to that on symbolic regression
* Appendix G - added; illustrative qualitative evaluation with GPT-3.
* Appendix H - moved from above and updated; the datasheet for datasets
* Appendix I - the author statement

---

### Meta-Review · Area_Chair_mxi3 · 2022-09-08

**Recommendation:** Accept
**Confidence:** 4

**Metareview:**

This paper presents a large dataset of integer sequences and several tasks for evaluating modeling of those sequences.

The reviewers generally appreciated the careful curation of the dataset, the range of models explored, and the potential application domains.

GVg5 and GeCe also bring up a point about human performance, which I also interpret as asking about the identifiability of these sequences similar to a point raised by Bx7J. I am convinced by the authors' response to this point.

I agree with reviewer TiHT that it is disappointing not to see more engagement with symbolic understanding of these sequences. Figure 1 seems to set up an approach for dealing with these sequences via natural language descriptors of rules or abstractions, but this is not the problem being addressed in this paper. While I think this would be interesting to see covered, the work doesn't necessarily need to focus on it.

Overall, this paper is very thoroughly done and appears to substantiate its claims.

---

### Decision · Program_Chairs · 2022-09-16

Accept